# Transarterial Radioembolization for Unresectable Hepatocellular Carcinoma: Real-Life Efficacy and Safety Analysis of Korean Patients

**DOI:** 10.3390/cancers14020385

**Published:** 2022-01-13

**Authors:** Sun Young Yim, Ho Soo Chun, Jae Seung Lee, Ji-Hwan Lim, Tae Hyung Kim, Beom Kyung Kim, Seung Up Kim, Jun Yong Park, Sang Hoon Ahn, Gyoung Min Kim, Jong Yun Won, Yeon Seok Seo, Yun Hwan Kim, Soon Ho Um, Do Young Kim

**Affiliations:** 1Department of Internal Medicine, Korea University College of Medicine, Korea University Anam Hospital, Seoul 02841, Korea; eug203@korea.ac.kr (S.Y.Y.); takers@korea.ac.kr (J.-H.L.); lacid@korea.ac.kr (T.H.K.); drseo@korea.ac.kr (Y.S.S.); umsh@korea.ac.kr (S.H.U.); 2Ewha Womans Medical Center, Department of Internal Medicine, Ewha Womans University College of Medicine, Seoul 03760, Korea; lakesy@yuhs.ac; 3Department of Internal Medicine, Yonsei University College of Medicine, Seoul 03722, Korea; SIKARUE@yuhs.ac (J.S.L.); BEOMKKIM@yuhs.ac (B.K.K.); KSUKOREA@yuhs.ac (S.U.K.); DRPJY@yuhs.ac (J.Y.P.); ahnsh@yuhs.ac (S.H.A.); 4Institute of Gastroenterology, Yonsei University College of Medicine, Seoul 03722, Korea; 5Yonsei Liver Center, Severance Hospital, Yonsei University Health System, Seoul 03722, Korea; 6Department of Radiology, Severance Hospital, Research Institute of Radiological Science, Yonsei University School of Medicine, Seoul 03722, Korea; gyoungmin@yuhs.ac (G.M.K.); JYWON@yuhs.ac (J.Y.W.); 7Department of Radiology, Korea University Anam Hospital, Seoul 02841, Korea; yhkku@kumc.or.kr

**Keywords:** hepatocellular carcinoma, overall survival, progression-free survival, risk factor, transarterial chemoembolization

## Abstract

**Simple Summary:**

While guidelines endorse locoregional intra-arterial therapies for intermediate-stage hepatocellular carcinoma (HCC) and systemic chemotherapies for advanced-stage HCC, there is emerging literature on transarterial radioembolization (TARE) with 90Y radioembolization. Our present study included a large number of patients from two major hospitals in Korea that may represent real-life efficacy data of unresectable HCCs. This study comprised 24% of Barcelona Clinic Liver Cancer (BCLC) A, 42% of BCLC B and 34% of BCLC C staged HCCs, which may represent real-life efficacy data across BCLC stages. The best overall tumor response, median overall survival and progression-free survival were comparable or even better than previous studies, especially for the intermediate-stage. Furthermore, the toxicities were durable, patients recovered without complication and none experienced adverse effects from the irradiation of non-target tissues.

**Abstract:**

Transarterial radioembolization (TARE) has become widely used in the treatment of HCC, one of the most common causes of cancer mortality worldwide. Here we investigated the long-term clinical outcomes of patients with hepatocellular carcinoma (HCC) treated with TARE in a multi-medical center in Korea. A total of 149 patients treated with TARE from 2008–2014 were recruited. The pre-treatment HCC stage was classified according to the BCLC stage, of which C and D were defined as advanced HCC. Advanced HCC stage and Child–Turcotte–Pugh (CTP) score A were identified in 62 (42%) and 134 (90%) patients, respectively. Portal vein thrombosis (PVT) was identified in 58 patients (38.9%). The median time to progression (TTP) was 14 months, and the median overall survival (OS) and progression-free survival (PFS) were 18.6 and 8.9 months, respectively. The overall tumor response was 47%, and the disease control rate was 78%. OS and PFS differed significantly according to the presence of liver cirrhosis, extrahepatic metastasis, tumor response and curative treatment after TARE (all, *p* < 0.05). Multiple tumors and major PVT were other independent factors related to OS, while the des-gamma carboxy protein level predicted PFS (all, *p* < 0.05). Tumor size was an independent predictor of tumor response. TTP, OS and PFS all differed among BCLC stages. The serious adverse effect after TARE was clinically not significant. Therefore, TARE is safe and effective in treating early to advanced HCCs.

## 1. Introduction

Hepatocellular carcinoma (HCC), the most common primary liver malignancy, is a leading cause of cancer-related death worldwide [1,2]. The crude annual rate of HCC mortality has increased over the last 30 years in Korea [3]. The initial approach in the management of HCC is to determine whether surgical resection or liver transplantation is possible [2]. However, many cases of HCC are diagnosed at intermediate or advanced stages and, thus, not candidates for curative treatment [4,5]. Transarterial chemoembolization (TACE) is used to treat non-curable HCC and plays a palliative role in selected patients [6,7]. In addition, many treatments, including systemic chemotherapy, molecular target agents and immune checkpoint inhibitors, have been used to improve the survival of rather than cure patients [8,9].

Transarterial radioembolization (TARE) was recently presented as a new modality of radiation therapy for HCC [10,11,12]. TARE is a liver-directed localized internal radiation therapy and microembolic procedure for HCC [13]. In TARE, microspheres impregnated with radioisotope yttrium 90 (90Y) are delivered through the hepatic artery to tumors with preferential blood flow [13]. A previous study reported that TARE for patients with advanced HCC is safe and effective and can be utilized even in patients with compromised liver function [11]. Another study demonstrated the safety and clinical benefit of TARE in patients with unresectable HCC complicated by portal vein thrombosis (PVT) and showed no increased risk of hepatic failure or encephalopathy in patients with branch or no PVT versus main PVT [12].

TARE could be an alternative to repeated TACE for patients who fail to respond to initial TACE or the first option for patients who are poor candidates for TACE [14]. Several previous studies showed that overall efficacy and time to progression are similar between the two modalities but that TARE features superior safety [15]. Liver resection is not recommended for intermediate-stage HCC according to the BCLC classification, but in recent decades, advances in surgical techniques have expanded indication for liver resection, including for intermediate-stage HCCs. There are many studies reporting improved survival in patients treated with resection over TACE in terms of survival benefit, and early resection could avoid TACE refractoriness [16,17,18,19,20]. However, for those who are not suitable for resection due to portal hypertension, TARE could be another treatment option. In addition, TARE can be used by itself to achieve a complete cure or as a safe bridge to resection that provides future liver remnant hypertrophy and disease control [21,22]. Despite previous studies that favor TARE, long-term outcome data regarding its efficacy and toxicity in Korea are insufficient.

Therefore, we investigated the long-term clinical outcomes of a multicenter cohort of HCC patients who were treated with TARE in Korea.

## 2. Materials and Methods

### 2.1. Patients

HCC patients who were treated with TARE between December 2008 and August 2014 at Severance Hospital and Korea University Anam Hospital, Seoul, Korea, were enrolled in this retrospective cohort study. The inclusion criteria were patients with clinical or histological diagnosis of HCC according to HCC guidelines [3], Child–Pugh class A/B, bilirubin < 2.5 mg/dL and AST/ALT < 5× upper limit of normal. The exclusion criteria were as follows: (1) age < 19 years, (2) Eastern Cooperative Oncology Group Performance Status (ECOG) score ≥ 3; (3) decompensated liver cirrhosis; (4) gastrointestinal uptake and lung shunt; (5) mortality due to other cancers (Figure 1).

The study protocol adhered to the ethical guidelines of the 1975 Declaration of Helsinki and was approved by the institutional review board of each participating institution. Informed consent was not required because of the retrospective study design.

### 2.2. Pretreatment Assessment

HCC was diagnosed histologically or clinically according to the guidelines proposed by the Korea Liver Cancer Study group, and a clinical diagnosis was based on a positive radiologic finding for typical HCC on dynamic computed tomography (CT) or magnetic resonance imaging (MRI) described as increased arterial enhancement followed by decreased enhancement compared to the liver (washout) in the portal or equilibrium phase. Portal vein thrombosis (PVT) of tumor origin was defined at baseline CT or MRI as a filling defect, partially or completely occluding the vessel in the portal venous phase with enhancement during the arterial phase of dynamic imaging [3].

HCC was staged by a Barcelona Clinic Liver Cancer (BCLC) stage (A, early; B, intermediate; C, advanced; D, end-stage) [23]. Liver function was also determined using Child–Turcotte–Pugh (CTP) and Eastern Cooperative Oncology Group (ECOG) performance status scores [24,25].

### 2.3. Treatment Modality

Pretreatment angioscintigraphy with technetium-99 macroaggregated albumin scanning was performed in all patients, and the lung shunt fraction was assessed to avoid the extrahepatic spread of microspheres labeled with 90Y. Two to three weeks after pretreatment, TARE was performed with resin (SIR-Spheres^®^; Sirtex Medical, Sydney, Australia) or glass particles (Thera-Sphere^®^; Biocompatibles UK Ltd., Surrey, UK) loaded with 90Y. The dose was determined from the planning angiogram and prepared in the nuclear medicine department according to the manufacturer’s preparation guide. The target-absorbed radiation doses to the normal liver and lungs should not exceed 70 and 25 Gy, respectively. TARE was conducted as previously described [26].

### 2.4. Follow-Up and Assessment of Treatment Responses

Approximately 1 and 3 months after the completion of TARE, patients underwent a contrast-enhanced CT or MRI to evaluate their treatment responses, after which imaging follow-up was performed at 2- or 3-month intervals.

The modified Response Evaluation Criteria In Solid Tumors (mRECIST) guidelines were used for the response evaluations [27]. For target lesions, the tumor response was defined as a complete response (CR), indicated by the complete disappearance of viable lesions, or as a partial response (PR), defined as at least a 30% decrease in the sum of diameters of viable target lesions, taking as reference the baseline sum of the diameters of target lesions. Progressive disease (PD) was defined as an increase of at least 20% in the sum of the diameters of viable target lesions, taking as reference the smallest sum of the diameters of viable target lesions recorded since treatment started. Stable disease was any case that did not qualify for either PR or PD. Objective response was defined as the proportion of participants whose best overall response is either CR or PR. Disease control was defined as the proportion of participants whose best overall response is CR, PR or SD. Responders were defined as the sum of the patients who showed an objective response.

Adverse events (AEs) were classified according to the Common Terminology Criteria for Adverse Events, version 4.0. Liver function toxicity within 3 months following treatment was assessed, and only those cases possibly related to the procedure were monitored over 12 months.

After progression following TARE, patients were treated according to clinician judgement or received the best supportive care. Even if progression or recurrence occurred, all enrolled patients were followed-up until death.

### 2.5. Statistical Analyses

Data are expressed as mean ± standard deviation, median (range) or *n* (%), as appropriate. Differences between continuous and categorical variables were examined for statistical significance using a Student’s t-test (or the Mann–Whitney test, as appropriate) and the chi-squared test (or Fisher’s exact test, as appropriate). Paired related continuous variables were analyzed using a paired t-test (or Wilcoxon paired test, as appropriate).

Time to progression (TTP) was calculated from the first TARE to the first progression at any site. Overall survival (OS) was analyzed from the first TARE to death from any cause, and progression-free survival (PFS) was calculated from the first TARE to the first progression at any site or death. OS with censoring to liver transplantation was assessed; OS and PFS were calculated using the Kaplan–Meier method and compared using the log-rank test.

Multivariate analysis was performed by adjusting for significant variables (*p* < 0.05) in the univariate analyses. The Cox proportional hazards model was used to perform the multivariate analysis. For a multivariate analysis using composite variables, the constituent variables were not incorporated into the multivariate analysis to prevent statistical collinearity. The adjusted hazard ratio and 95% confidence interval were calculated for each selected risk factor.

All statistical analyses were performed using the Statistical Package for Social Sciences (SPSS version 25.0; IBM, Armonk, NY, USA). Statistical significance was set at *p* < 0.05.

## 3. Results

### 3.1. Patient, Tumor, and Treatment Characteristics

The baseline clinical characteristics of the 149 patients are presented in Table 1. The median follow-up duration was 16.9 (range: 1–139 months) months. The mean age was 60.6 ± 12.6 years, and 117 patients were male (78.5%). The most common cause of liver disease was hepatitis B viral infection (58.4%), followed by hepatitis C viral infection (28.2%) and alcohol (6.7%). The majority (90.6%) of the patients were treatment-naïve, while 12 (8.1%) patients had previously undergone TACE and 2 (1.3%) patients had resectioned before TARE (Table 1).

Baseline liver function was well preserved with a mean serum bilirubin level of 0.8 ± 0.5 mg/dL and a serum albumin level of 3.7 ± 0.5 g/dL. Ascites was observed in 15 patients. Lymph node, solid organ or bone metastasis was observed in 27 (18.1%) patients. Although main PVT was observed in only 5 (3.4%) patients, either right or left PVT was observed in 37 (24.8%) patients, and segmental branch occlusion was observed in 16 (10.7%) patients. Hepatic vein and bile duct invasion were observed in 8.7% and 6.7% of patients, respectively. Most of the patients were classified as BCLC stage C (41.6%), followed by B (34.2%) and A (24.2%).

The mean tumor burden was 18.1 ± 16.3%, while hepato-pulmonary shunt was 10.3 ± 6.2%. More than half of the patients (56.4%) received SIR-spheres targeting the right lobe, followed by the whole liver (25.5%), and one or more segments (12.8%), whereas the left lobe was targeted in only eight patients (Appendix A). The mean administered dose was 1.98 ± 0.89 GBq, and the mean radiation dose absorbed by the tumor was 235 ± 195 Gy.

Seventeen patients (11.4%) had curative (resection or liver transplantation) treatment, and 28 patients (18.8%) received systemic treatment such as sorafenib and radiation therapy, while the majority of the patients (41.6%) had repeated TACE after TARE.

### 3.2. Radiological Tumor Response

Among the 149 patients, tumor responses were evaluated according to the mRECIST guidelines. The best overall response at any time during follow-up was analyzed in total patients as well as according to BCLC stages (Table 2). Objective response was observed in 67 (45.9%) patients, while disease control rate was observed in 74.5% of evaluated patients. The objective response and disease control rates were significantly lower in BCLC stage C patients compared to BCLC A and B (both, *p* < 0.001).

During the study follow-up, 86 progressions were observed. The median TTP for the entire cohort was 14.3 months (range: 1–139 months) with significant differences according to BCLC stages. For BCLC A, median TTP was 42.5 months followed by 12.7 months for BCLC B (*p* = 0.082) and BCLC C 8.9 months (*p* = 0.028) (Appendix A).

### 3.3. Overall Survival and Progression-Free Survival

During the follow-up, 119 (79.9%) deaths were observed, and among them, tumor progression was observed in 73 (61.3%) patients before death. No deaths were observed within one month of treatment. Among 119 deaths, HCC-related death was observed in 86 (71.4%) patients, liver failure (e.g., esophageal bleeding, hepatorenal syndrome bleeding) in 7 (5.9%) patients, while non-liver related death due to sepsis occurred in 8 (6.7%) patients. The causes of 18 deaths were not specified due to follow-up loss, but death was identified using the national health insurance data. The median OS and PFS were 18.6 months (range: 1–139 months) and 8.9 months (range: 0.7–139 months), respectively (Figure 2a,b).

OS rates at 1, 2, 3 and 5 years were 62.1%, 45.2%, 36.2% and 30.9%, respectively, while PFS rates were 42.4%, 31.2%, 22.7% and 17.7%, respectively. OS and PFS rates were stratified according to CTP score and the presence of major PVT. Median OS was significantly better for patients classified as CTP A than B (21.2 vs. 5 months, *p* < 0.001) and significantly better for those without PVT than those with PVT (24.4 vs. 7 months, *p* = 0.003) (Figure 3a,b). Reflecting the influence of liver function and tumor aggressiveness, the median survival of BCLC A was 51.3 months which decreased to 27.5 months for BCLC stage B and decreased significantly to 6.5 months for BCLC stage C (Figure 3c).

Similar results were observed for PFS, with better median values for patients with CTP A than CTP B (9.3 vs. 3.7 months, *p* < 0.001), for those without PVT than for those with PVT (11.8 vs. 4.9 months, *p* = 0.002) and those with BCLC A than those with BCLC B (25.7 vs. 12 months, *p* = 0.056) followed by those with BCLC C (4.6 months, *p* < 0.001) (Figure 4a–c).

In addition to baseline characteristics, tumor control rate significantly affected the patient outcome. In responders (CR and PR) versus non responders, a significant difference was observed for median values of OS (41.9 vs. 11.6 months, *p* < 0.001) and PFS (22 vs. 4.6 months, *p* < 0.001) (Figure 5a,b).

The prognoses of the patients were also analyzed based on different types of treatment modalities following TARE (Table 1). The median OS for curative treatment was not reached while TACE was 32.3 months and systemic/radiation therapy was 4.4 months, *p* < 0.001. The median PFS for curative treatment was 27.5 months followed by 14.4 months with TACE and 4 months with systemic/radiation therapy, all were *p* < 0.05. Best supportive care led to similar OS and PFS with systemic/radiation therapy (Appendix A).

### 3.4. Prognostic Factors

The univariate analysis results for factors associated with OS and PFS are shown in Table 3 and Table 4. The multivariate analysis results for independent factors associated with OS were the presence of liver cirrhosis, multiple tumors, extrahepatic metastasis, major portal vein thrombosis, objective response and post-TARE treatments. When BCLC was included for analysis, worse performance status, increasing BCLC stage, non-responders and non-curative treatment after TARE were independently associated with lower survival.

For PFS, multivariate analysis revealed that BCLC stage, presence of liver cirrhosis, increased DCP level, presence of an objective response and types of treatments predicted PFS independently (Table 4).

### 3.5. Predictive Factors for Achieving Tumor Response

Since responders were significantly associated with better OS and PFS, identification of factors that predict tumor response are important in clinical settings. Univariate analysis revealed that DCP level (*p* = 0.009), larger tumor size (<0.001), tumor burden (*p* = 0.019), PVT including segmental branching (*p* = 0.013), extrahepatic metastasis (*p* = 0.012) and non-curative treatment (*p* = 0.008) were related to non-response.

Subsequent multivariate analysis revealed that tumor size was the most important predictor of response (*p* = 0.022) while metastasis and non-curative treatment were likely to predict tumor response, *p* = 0.075 and 0.074, respectively.

### 3.6. Toxicity after TARE

Complications occurred in 60 patients (40.3%); among them, grade 3 adverse events (AEs) occurred in four within 7 days after TARE. Since patients could have more than one AE, the AEs were (123 total) were stratified according to toxicity grade (Table 5). Grade 1/2 AEs accounted for 91.1% of the total AEs, while grade 3 AEs accounted for the other 8.9%. The most common grade 1/2 AEs were gastrointestinal symptoms, such as abdominal pain, nausea, anorexia, fatigue and diarrhea. Grade 3 AEs included nausea, anorexia, abdominal pain that improved without sequelae and other AEs (cholecystitis, sepsis, celiac trunk dissection and HCC rupture). Serious AEs did not lead to death; rather, the cause of death was HCC progression. Liver function toxicity was observed in 14 patients at 3 months of treatment, including 6 with grade 3 toxicity.

## 4. Discussion

The clinical value of TARE with 90Y microspheres for the management of advanced HCC remains under debate. However, a considerable amount of information has been published in the last few decades, and growing evidence supports the role of radioembolization as a safe and efficient treatment for unresectable HCC. Our study showed that the median OS was 18.6 months with a 3-year survival rate of 36%. Our real-life data results are promising, and our results are in concordance with those of previous studies. Salem et al. [28] reported a median OS of 17.4 months in 123 patients, Mantry et al. [29] reported 13.1 months in 115 patients and Thai et al. [30] reported a median survival of 23.9 months in 97 patients with a 3-year survival rate of 31%. The value of radioembolization became more prominent, as the median OS of unresectable HCC cases treated with sorafenib in the SHARP trial was 10.7 months [31] versus 13.6 months for those treated with lenvatinib in a phase 3 trial [32].

Two recent phase 3 trials—one in Europe (SARAH) [33], the other in the Asia-Pacific region (SIRveNIB) [34]—showed that despite the fact that TARE failed to show improved OS versus sorafenib, its use was associated with a higher response rate, longer time to progression and a lower AE rate. The median PFS was 4.1 months in the SARAH trial and 5.8 months in the SIRveNIB trial. Since the trials included mostly locally advanced HCC cases and our study comprised a small number of early-stage HCC cases, a further subgroup analysis was performed for BCLC B and C cases only, which revealed similar results, with a median PFS of 6.1 months, indicating that TARE could be an alternative treatment to sorafenib in advanced HCC cases.

The benefit of TARE versus TACE in terms of TTP was reported by Selem et al. [28], who demonstrated an overall median TTP of 8.4 months for conventional TACE and 13.3 months for TARE. Considering that the median TTP of TARE-treated patients ranged from 8 to 11 months in previous studies, our study with a median TTP of 14.3 months for all patients and 12.7 months for the intermediate stage were comparable to that of previous studies indicating that TARE could be another treatment option for HCC cases indicated for TACE.

The efficacy of TARE was again confirmed by an overall response of 45% and DCR above 75%. The tumor response rate for BCLC B/C was similar to the study by Mazzaferro et al. (ORR 40.4%, DCR 78.8%) [4] while better than those reported in SIRveNIB [34] (ORR 41.8%, DCR 58.8%) and the SARAH trial (ORR 19%, DCR 68%) [33]. Considering the dismal prognosis of unresectable HCC cases, with overall response and disease control rates of 50–70% and 70–90%, respectively, in TACE-treated patients, it is noteworthy that TARE could be an alternative treatment, especially in cases of huge HCC with PVT or hepatic vein invasion, since TACE is relatively contraindicated in such patients [35,36]. Furthermore, TARE as a bridge treatment before liver transplantation was successful in nine patients, including two patients with major PVT at the time of TARE, after reaching overall response.

The effect of tumor response was significantly related to patient outcome, both OS and PFS, and it is noteworthy that a smaller tumor size was associated with responders. We also identified that liver cirrhosis and extrahepatic metastasis had a negative association with OS and PFS. Disease factors such as CTP score, multiple tumors and major PVT predicted those who will benefit most from TARE while high DCP levels predicted PFS. Down-staging following TARE that eventually enabled curative treatment such as resection and liver transplantation was also another important factor that predicted both OS and PFS.

The above findings might explain easily as multiple tumors have a higher possibility of micro-vessel invasion while the presence of metastasis disables TACE or curative treatment following TARE that all lead to non-response of HCC. This might indicate that if we select optimal candidates for TARE, a favorable long-term prognosis can be anticipated, especially in those who received curative treatment after TARE.

In most cases of the present study, TARE was indicated as the initial treatment for HCC instead of TACE or surgical resection. The patient and tumor factors (age, ECOG, tumor size and major PVT) were considered before performing TARE. Among BCLC A patients, 8 patients had a single HCC with a tumor size <5 cm, 25 patients had a single HCC sized 5–10 cm and 1 patient had a single HCC measuring >10 cm. Resection could be another treatment option, but TARE was performed as 71% of these patients were older than 60 years old, and 76% of these patients were older than 75 years. The presence of portal hypertension (mild ascites and esophageal varices) and patients’ preference were other reasons. There are studies reporting similar outcomes of TARE to those of ablation and surgical resection [37,38]. Vouche et al. assessed the efficacy of radiation segmentectomy in solitary HCC ≤ 5 cm not amenable to ablation, reporting a median TTP of 33.1 months and median OS of 53.4 months [37]. These results are comparable to our result of BCLC A with a median TTP of 43 months and median OS of 51.3 months.

Furthermore, TARE was chosen as the first treatment option for those with multiple tumors with major PVT as it is reported to show better performance than TACE, low incidence of complications and persistence of response after a single treatment which are attractive features of TARE.

In particular, no treatment-related deaths were observed. The advantage of radioembolization compared to current standard therapies such as TACE and sorafenib is its tolerable toxicity. Only four patients suffered grade ≥3 toxicity, most cases of which were related to post-embolization syndrome that improved without sequelae. According to the SARAH trial, quality of life and safety were better in the TARE group than in the sorafenib group [33]. A dose reduction is often warranted in patients who are treated with sorafenib, which in turn may have a detrimental effect on treatment efficacy [39,40]. A systematic review that compared TARE to standard treatments such as sorafenib and TACE also showed that grade ≥3 toxicity was significantly less common in the TARE-treated group than in patients treated with standard treatment [41]. The known AEs for TARE, the irradiation of non-target tissue rather than from embolic effects such as gastrointestinal tract ulcers, pneumonitis and radiation-induced liver injury, were not observed in our study population.

The results of our study are limited by its retrospective nature, especially when grading AEs, as CTCAE grades were not routinely assessed. Furthermore, the various treatments following TARE could have affected the survival analysis.

However, the results from the current study were based on a relatively large number of patients with a fairly long-term follow-up, adding value to the growing literature on TARE, demonstrating that it is a safe and effective treatment modality for patients with intermediate-or advanced-stage HCC.

## 5. Conclusions

TARE is safe and effective for patients from early to advanced HCC with a high tumor response of 45% in overall patients and even in BCLC B/C (40%) with a median OS 18.6 months. The absence of liver cirrhosis, presence of tumor response and curative treatment following TARE are predictors of both OS and PFS, while tumor size independently predicted tumor response. Our findings could be of help in treatment guidance and may provide evidence to consider TARE as a treatment option for early to advanced HCCs

## Figures and Tables

**Figure 1 cancers-14-00385-f001:**
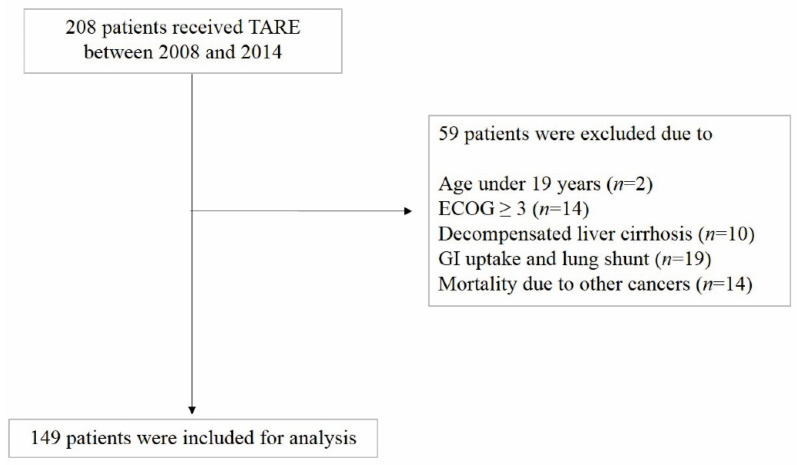
Flow chart of the patient selection process.

**Figure 2 cancers-14-00385-f002:**
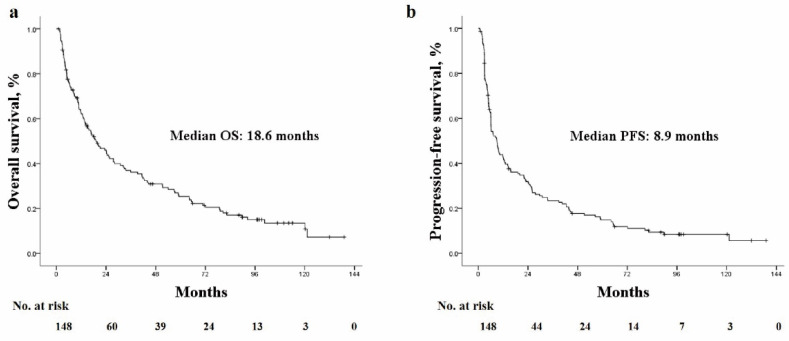
The Kaplan–Meier curves of (**a**) OS and (**b**) PFS. OS, overall survival; PFS, progression-free survival.

**Figure 3 cancers-14-00385-f003:**
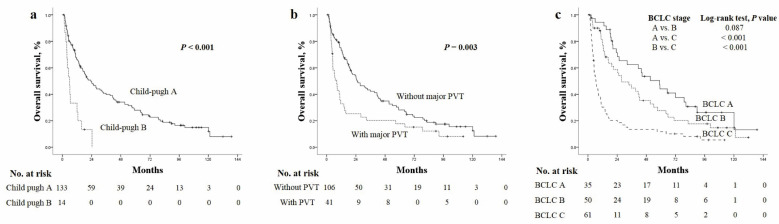
The Kaplan–Meier curves of overall survival according to (**a**) Child–Pugh class, (**b**) presence of major PVT and (**c**) BCLC stages.

**Figure 4 cancers-14-00385-f004:**
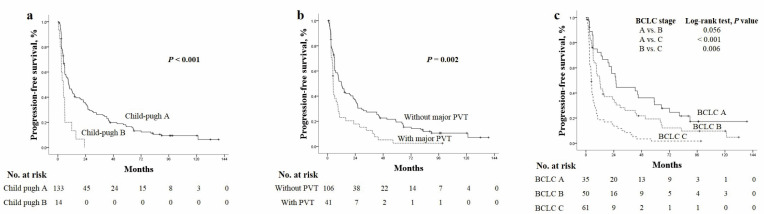
The Kaplan–Meier curves of overall survival according to (**a**) Child–Turcotte–Pugh score, (**b**) presence of PVT and (**c**) BCLC stages. BCLC, Barcelona Clinic Liver Cancer; HCC, hepatocellular carcinoma; PVT, portal vein thrombosis.

**Figure 5 cancers-14-00385-f005:**
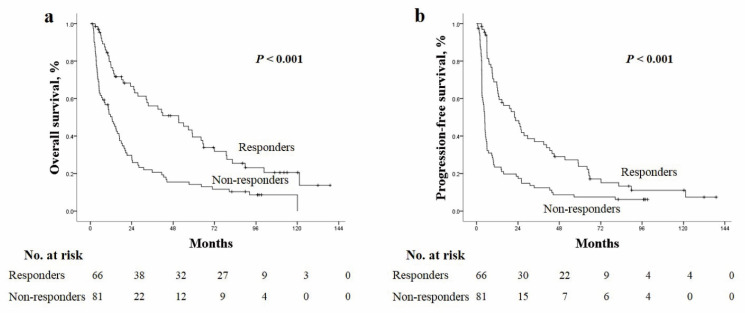
The Kaplan–Meier curves of (**a**) overall survival and (**b**) progression-free survival according to the presence of responders (complete and partial response).

**Table 1 cancers-14-00385-t001:** Baseline characteristics of the study population.

Variables	Patients, *N* = 149
**Demographic variables**	
** Age (y)**	60.6 ± 12.6
** Male sex**	117 (78.5)
** Etiology**	
Hepatitis B virus	87 (58.4)
Hepatitis C virus	42 (28.2)
Alcohol	10 (6.7)
Cryptogenic	7 (4.7)
Hepatitis B virus and hepatitis C virus	2 (1.3)
Hepatitis C virus and alcohol	1 (0.7)
** Previous HCC therapy**	
None	135 (90.6)
Resection	2 (1.3)
TACE	12 (8.1)
** Liver cirrhosis**	94 (63.1)
** Ascites**	15 (10.1)
** Child–Turcotte–Pugh class**	
A	134 (89.9)
B	15 (10.1)
** ECOG score**	
0	63 (42.3)
1	75 (50.3)
2	11 (7.4)
**Laboratory variables**	
Total bilirubin (mg/dL)	0.8 ± 0.5
Serum albumin (g/dL)	3.7 ± 0.5
Prothrombin time (INR)	1.06 ±0.12
AFP (ng/mL)	19,884 ± 93,841
>200	72 (48.3)
DCP (mAU/mL)	12,428 ± 23,879
>15,000	27 (18.1)
**Tumor characteristics**	
** Tumor number**	
1	52 (34.9)
2–3	42 (28.2)
>3	55 (36.9)
** Largest tumor diameter, cm**	7.7 ± 3.5
** BCLC stage**	
A	36 (24.2)
B	51 (34.2)
C	62 (41.6)
** Metastasis**	
None	122 (81.9)
Lymph node	7 (4.7)
Solid organ	14 (9.4)
Bone	6 (4)
** Tumor location**	
Bilobar	66 (44.3)
Unilobar	83 (55.7)
** Tumor burden > 25%**	41 (27.5%)
** Tumor-related PVT**	
None	91 (61.1)
Segmental branch	16 (10.7)
Rt. or Lt. portal vein	37 (24.8)
Main portal vein	5 (3.4)
** Hepatic vein invasion**	13 (8.7)
** Bile duct invasion**	10 (6.7)
**Treatment modalities after TARE**	
Resection	8 (5.4)
Liver transplantation	9 (6)
TACE	62 (41.6)
Radiotherapy and systemic treatment	28 (18.8)
None	42 (28.2)
Best supportive care	33
Not evaluable	9

Variables are expressed as mean ± SD or *n* (%); AFP, alpha-fetoprotein; BCLC, Barcelona Clinic Liver Cancer; CI, confidence interval; DCP, des-gamma carboxy protein; ECOG, Eastern Cooperative Oncology Group Performance Status; HCC, hepatocellular carcinoma; HR, hazard ratio; INR, international normalized ratio; PVT, Portal Vein Thrombosis; TACE, transarterial chemoembolization.

**Table 2 cancers-14-00385-t002:** Best overall response of TARE.

Overall Radiological Response (From Any Time on Study)	Total Patients (*n*, %)	BCLC A (*n*, %)	BCLC B (*n*, %)	BCLC C (*n*, %)
Number of patients	149	36	51	62
Complete response	23 (15.4)	10 (27.8)	11 (21.6)	2 (3.2)
Partial response	44 (29.5)	12 (33.3)	19 (37.3)	13 (21)
Stable disease	44 (29.5)	10 (27.8)	16 (31.4)	18 (29)
Progressive disease	38 (25.5)	4 (11.1)	5 (9.8)	29 (46.8)
Objective response (CR + PR)	67 (45)	22 (61.1)	30 (58.8)	15 (24.2)
Disease control (CR + PR + SD)	111 (74.5)	32 (88.9)	46 (90.2)	33 (53.2)

RECIST, Response Evaluation Criteria in Solid Tumors; TARE, transarterial chemoembolization.

**Table 3 cancers-14-00385-t003:** Factors contributing to overall survival.

Variable	Overall Survival								
	Univariate Analysis			Multivariate Analysis			Multivariate Analysis Using BCLC Stage		
	HR	95% CI	*p* Value	HR	95% CI	*p* Value	HR	95% CI	*p* Value
**Demographic variables**									
Age (≥60 y)	1.001	0.986–1.001	0.725	-	-	-	-	-	-
Male sex	1.027	0.655–1.611	0.906	-	-	-	-	-	-
Etiology (CHB)	1.414	0.985–2.030	0.06	-	-	-	-	-	-
Previous HCC therapy	2.147	1.165–3.959	0.014	1.487	0.633–3.494	0.362	0.896	0.418–1.924	0.779
Liver cirrhosis	1.55	1.054–2.279	0.026	1.996	1.28–3.112	0.009	1.393	0.882–2.199	0.155
Ascites	1.686	0.925–3.073	0.088	-	-	-	-	-	-
CTP score									
A	1	-	-	-	-	-	-	-	-
B	3.507	1.949–6.311	<0.001	1.81	0.845–3.876	0.143	-	-	-
ECOG									
0	1	-	<0.001	1	-	0.428	1	-	0.019
1	1.727	1.777–2.534	0.005	1.326	0.863–2.037	0.197	1.813	1.199–2.74	0.005
2	3.746	1.868–7.511	<0.001	1.261	0.505–3.147	0.62	1.323	0.568–3.082	0.517
**Laboratory variables**									
Bilirubin (mg/dL)	1.989	1.429–2.769	<0.001	-	-	-	-	-	-
Albumin (g/dL)	0.45	0.324–0.625	<0.001	-	-	-	-	-	-
PT (INR)	4.905	0.867–27.732	0.072	-	-	-	-	-	-
AFP > 200 ng/mL	1.496	1.037	0.031	0.942	0.607–1.464	0.791	0.944	0.615–1.45	0.794
DCP > 15,000 mAU/mL	1.538	1.05–2.251	0.027	1.307	0.716–2.385	0.383	1.357	0.804–2.29	0.253
**Tumor characteristics**									
Size ≥ 10 cm	1.821	1.19–2.787	0.006	0.663	0.326–1.349	0.257			
Tumor number >3	2.438	1.674–3.552	<0.001	2.457	1.552–3.888	<0.001			
BCLC stage									
A	1	-	<0.001	-	-	-	1	-	<0.001
B	1.444	0.87–2.397	0.155	-	-	-	2.278	1.279–4.059	0.005
C	3.498	2.164–5.654	<0.001	-	-	-	3.925	2.238–6.886	<0.001
Presence of metastasis	5.683	3.482–9.275	<0.001	2.773	1.571–4.895	0.004			
Bilobar location	1.424	0.992–2.044	0.055	-	-	-	-	-	-
Tumor burden >25%	1.579	1.068–2.336	0.022	1.426	0.803–2.532	0.226	0.897	0.55–1.463	0.664
Portal vein thrombosis	1.796	1.2096–2.668	0.004	1.831	1.172–2.858	0.008	-	-	-
Hepatic vein invasion	5.102	2.786–9.344	<0.001	1.855	0.943–3.65	0.074	-	-	-
Bile duct invasion	1.451	0.734–2.869	0.284	-	-	-	-	-	-
**Objective response**	0.437	0.3–0.636	<0.001	0.424	0.271–0.663	<0.001	0.486	0.316–0.747	<0.001
**Post TARE treatment**									
Curative			<0.001	1	-	<0.001	-	-	<0.001
TACE	15.801	2.183–114.381	0.006	13.55	1.834–100.133	0.01	20.51	2.81–149.65	0.003
RTx/Systemic CTx/BSC	53.702	7.232–398.797	<0.001	32.35	4.347–240.833	0.001	41.72	5.55–313.55	<0.001

AFP, alpha-fetoprotein; BCLC, Barcelona Clinic Liver Cancer; BSC, best supportive care; CI, confidence interval; CLIP, Cancer of the Liver Italian Program; CTP, Child–Turcotte–Pugh; CTx, chemotherapy; DCP, des-gamma carboxy protein; ECOG, Eastern Cooperative Oncology Group Performance Status; HCC, hepatocellular carcinoma; HR, hazard ratio; INR, international normalized ratio; PT, prothrombin time; RTx, radiation treatment; TACE, transarterial chemoembolization.

**Table 4 cancers-14-00385-t004:** Factors contributing to progression-free survival.

Variable	Progression-Free Survival								
	Univariate Analysis			Multivariate Analysis			Multivariate Analysis Using BCLC Stage		
	HR	95% CI	*p* Value	HR	95% CI	*p* Value	HR	95% CI	*p* Value
**Demographic variables**									
Age (≥60y)	0.991	0.702–1.397	0.957	-	-	-	-	-	-
Male sex	0.926	0.602–1.425	0.728	-	-	-	-	-	-
Etiology (CHB)	0.782	0.553–1.105	0.163	-	-	-	-	-	-
Previous HCC therapy	2.552	1.375–4.738	0.003	1.889	0.851–4.192	0.114	1.778	0.839–3.767	0.105
Liver cirrhosis	1.478	1.028–2.126	0.035	2.034	1.307–3.164	0.002	1.654	1.075–2.544	0.011
Ascites	1.26	0.709–2.24	0.43	-	-	-	-	-	-
CTP class									
A	1	-	-				-	-	-
B	2.684	1.542–4.671	<0.001	1.123	0.55–2.295	0.75	-	-	-
ECOG									
0	1	-	0.024	1	-	0.763	1	-	0.448
1	1.467	1.024–2.103	0.037	1.164	0.775–1.747	0.464	1.276	0.854–1.907	0.238
2	2.221	1.124–4.39	0.022	1.064	0.45–2.517	0.887	0.962	0.415–2.248	0.929
**Laboratory variables**									
Bilirubin (mg/dL)	1.684	1.203–2.357	0.002	-	-	-	-	-	-
Albumin (g/dL)	0.539	0.394–0.738	<0.001	-	-	-	-	-	-
PT (INR)	3.339	0.706–15.778	0.128	-	-	-	-	-	-
AFP > 200 ng/mL	1.434	1.013–2.029	0.042	0.9	0.593–1.366	0.62	0.902	0.605–1.344	0.611
DCP > 15,000 mAU/mL	2.308	1.483–3.592	<0.001	1.734	1.064–2.825	0.027	1.549	0.926–2.589	0.041
**Tumor characteristics**									
Size ≥ 10 cm	1.873	1.236–2.838	0.003	0.893	0.517–1.542	0.684	-	-	-
Tumor number (>3)	1.966	1.37–2.82	<0.001	1.355	0.833–2.205	0.221	-	-	-
BCLC stage									
A	1	-	<0.001	-	-	-	1	-	0.001
B	1.531	0.955–2.454	0.077	-	-	-	1.42	0.831–2.426	0.236
C	3.421	2.152–5.437	<0.001	-	-	-	2.544	1.518–4.261	<0.001
Presence of metastasis	3.705	2.344–5.857	<0.001	2.363	1.373-4.066	0.001	-	-	-
Bilobar location	1.177	0.834–1.66	0.353	-	-	-	-	-	-
Tumor burden >25%	1.529	1.046–2.234	0.028	1.197	0.691–2.074	0.521	0.946	0.604–1.481	0.807
Portal vein thrombosis	1.827	1.247–2.678	0.002	1.545	0.998–2.393	0.061	-	-	-
Hepatic vein invasion	3.365	1.865–6.069	<0.001	1.294	0.649–2.579	0.453	-	-	-
Bile duct invasion	1.86	0.971–3.562	0.061	-	-	-	-	-	-
**Objective response**				0.39	0.253–0.601	<0.001	0.46	0.298–0.709	<0.001
**Post TARE treatment**									
Curative	1	-	<0.001	1	-	0.016	-	-	0.007
TACE	15.801	2.183–114.381	0.006	2.024	0.949–4.317	0.097	2.311	1.093–4.888	0.063
RTx/Systemic CTx/BSC	53.702	7.232–398.797	<0.001	3.512	1.5–8.225	0.017	3.841	1.629–9.061	0.004

AFP, alpha-fetoprotein; BCLC, Barcelona Clinic Liver Cancer; BSC, best supportive care; CI, confidence interval; CLIP, Cancer of the Liver Italian Program; CTP, Child–Turcotte–Pugh; CTx, chemotherapy; DCP, des-gamma carboxy protein; ECOG, Eastern Cooperative Oncology Group Performance Status; HCC, hepatocellular carcinoma; HR, hazard ratio; INR, international normalized ratio; PT, prothrombin time; RTx, radiation treatment; TACE, transarterial chemoembolization.

**Table 5 cancers-14-00385-t005:** Clinical and laboratory toxicity after TARE graded by CTCAE v4.03.

Toxicity	Grade 1/2	Grade 3/4
	N (%)	N (%)
**Clinical toxicities**		
Nausea	21 (17.1)	1 (0.8)
Vomiting	3 (0.2)	0 (0.0)
Nausea and vomiting	1 (0.8)	0 (0.0)
Anorexia	13 (10.6)	1 (0.8)
Diarrhea	1 (0.8)	0 (0.0)
Weight loss	1 (0.8)	0 (0.0)
Abdominal pain	32 (26)	2 (1.6)
Fever	12 (7.3)	0 (0.0)
Fatigue	9 (7.3)	0 (0.0)
Cholecystitis	0 (0.0)	1 (0.8)
Gastritis	2 (1.6)	0 (0.0)
Other		
Pruritus	1 (0.8)	0 (0.0)
Splenic infarction	1 (0.8)	0 (0.0)
Post-procedural bleeding	1 (0.8)	0 (0.0)
Sepsis	0 (0.0)	1 (0.8)
Celiac trunk dissection	0 (0.0)	1 (0.8)
HCC rupture ^a^	0 (0.0)	1 (0.8)
**Laboratory toxicities**		
AST/ALT elevation	14 (11.4)	3 (2.4)

ALT, alanine aminotransferase; AST, aspartate aminotransferase; CTCAE, Common Terminology Criteria for Adverse Events; HCC, hepatocellular carcinoma; TARE, transarterial radioembolization; ^a^ HCC rupture occurred two weeks after the procedure.

## Data Availability

The data that support the findings of this study are available on request from the corresponding author D.Y.K.

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
