# Peer review of "Transarterial Radioembolization for Unresectable Hepatocellular Carcinoma: Real-Life Efficacy and Safety Analysis of Korean Patients"

_cancers, 2022, doi:10.3390/cancers14020385_

Round 1

Reviewer 1 Report

In this article, the authors present an interesting experience in the treatment of patients with unresectable hepatocarcinoma (HCC), using TARE. These experiences can contribute to a better definition of the indications and results of this treatment.

Although the article is well written and easy to read, I think that the authors should provide some data not included in the manuscript and respond to a few comments.

In most cases, TARE was indicated as the initial treatment for HCC. The authors should comment on the reasons why TACE was not indicated as the first treatment in this group of patients (contraindications,….).

In Table 1 it would be convenient to indicate if any of the portal venous thromboses were of tumor origin.

Did patients with extrahepatic disease receive systemic treatment? What was its evolution?

Do the authors have data on radiation dose absorbed by the tumor?

The causes of death should be specified.

The conclusions establish that TARE is safe and effective even in patients with compromised liver function. However, only 10.7% of the patients (n = 16) had a Child grade higher than A. On the other hand, median survival in Child B patients was only 4 months. Therefore, the number of cases with poor liver function included in the study and the results obtained do not clearly justify this conclusion.

The authors suggest that the study data may serve as a guide in the indication of TARE in the treatment of unresectable hepatocarcinomas. However, they do not make any specific recommendation as a result of their experience. For example, predictive factors associated with poorer survival  after TARE include previous treatment failure. Could this situation be a relative contraindication for the use of TARE?.

Author Response

  1. In most cases, TARE was indicated as the initial treatment for HCC. The authors should comment on the reasons why TACE was not indicated as the first treatment in this group of patients (contraindications,….).
  • Thank you for the comments. We agree that TACE has similar indications to TARE and TACE could be selected as first option of treatment. In our study only 8.1% had TACE prior to TARE while majority (90.6%) of the patients had TARE as first treatment option.
  • The reason for selecting TARE as first treatment option was due to large tumor size at the time of diagnosis. Among 149 patient, 108 patient had tumors measuring >5cm for the largest one and 26% of these patients had tumors >10cm.
  • Another reason for selecting TARE as first treatment option was due to multiple tumors with largest tumor size (<5cm) but with infiltrative morphology involving major portal vein (n=3).
  • TARE was also performed as a mean of radiosegmentectomy for tumors measuring <5cm without major vessel invasion (n=15).
  • Since TARE is known to cause less pain than TACE for larger tumors, has more evidence for HCCs with major PVT and and radiosegmentectomy has emerged as early treatment option TARE was performed with wider indication in our real life study.
  • In addition to the above mentioned reasons, approximately 9 patients eligible for both TACE and TARE were treated with TACE due to the patient preference.
  1. Table 1 it would be convenient to indicate if any of the portal venous thromboses were of tumor origin.
  • Thank you for the comment. All portal venous thrombosis were of tumor origin and were considered for BCLC staging system. This statement has been added in the table1.

  1. Did patients with extrahepatic disease receive systemic treatment? What was its evolution?
  • Metastasis was observed in 27 patients as stated in Table 1.
  • Among these patients, sorafenib was prescribed in 10 patients (8 lung; 2 lymph node metastasis) radiotherapy and sorafenib in 5 patients (4 bone, 1 aortocaval lymph node metastasis) and TACE in 5 patients 7 patients had best supportive care.
  • These patients showed median OS duration of 5 months and the cause of death was HCC progression in all patients.
  • This additional findings are mentioned in the discussion and treatments after TARE were evaluated in univariate and multivariate analysis to observe if there is association with OS and PFS.
  1. Do the authors have data on radiation dose absorbed by the tumor?
  • Yes we do have the data. The median radiation dose was 185Gy ranging from 73 to 1422 Gy.
  1. The causes of death should be specified.
  • Among 119 deaths, HCC-related death was observed in 86 (71.4%) patients, liver failure (eg esophageal bleeding, hepatorenal syndrome bleeding) in 7 (5.9%) patients while non-liver related death due to sepsis occurred in 8 (6.7%) patients. Nineteen deaths were not specified due to follow up loss but death was identified using the national health insurance data.
  • We have added these data under Results.
  1. The conclusions establish that TARE is safe and effective even in patients with compromised liver function. However, only 10.7% of the patients (n = 16) had a Child grade higher than A. On the other hand, median survival in Child B patients was only 4 months. Therefore, the number of cases with poor liver function included in the study and the results obtained do not clearly justify this conclusion.
  • Thank you for the comment. First child pugh C patient was classified as child pugh B in the revised manuscript. This patient was initially classified to child pugh C due to high bilirubin level observed after TACE which decreased to normal range at the time of TARE.
  • Although CTP was not an independent predictor of OS and PFS in multivariate analysis, the child pugh class B patients definitely showed lower OS and PFS in univariate analysis and therefore the conclusion have been modified based on our revised results

  1. The authors suggest that the study data may serve as a guide in the indication of TARE in the treatment of unresectable hepatocarcinomas. However, they do not make any specific recommendation as a result of their experience. For example, predictive factors associated with poorer survival after TARE include previous treatment failure. Could this situation be a relative contraindication for the use of TARE?.
  • Thank you for your critical comment. The results that could of help in clinical decision is important and should be emphasized. After adding several more factors such as tumor number, largest tumor diameters, presence of objective response and types of treatment following TARE diminished the effect of initial treatment failure. Detailed explanation for independent variables associated with survival are explained in the discussion.

Reviewer 2 Report

I have read with interest this manuscript that concerns the role of TARE in HCC patients. While the topic is of interest and the patients’ cohort is quite large, I believe that some changes should be made to clarify some issues and make the message more generalizable.

  1. All the abbreviations, starting from the abstract, should be spelled out.
  2. Not only the exclusion criteria but also the inclusion criteria should be detailed.
  3. Reading the introduction, it seems that apart from systemic therapy TAE or TARE are the only possible treatments for intermediate and advanced HCC patients. Please note that also surgery in selected cases of multinodular and/or macrovascular invasive HCC may be associated with adequate survival. The literature is replete of these data from surgical series, both from the Eastern and from the Western world.
  4. In this sense, TARE should be selected in those patients considered unresectable and MDT discussion should be warranted.
  5. I found interesting the use of different HCC classifications/guidelines: from the Korea Cancer Study Group, to the BCLC, to the CLIP. This is quite unusual. Please comment.
  6. How many patients treated with TARE were excluded because of technical issues, such as shunt?
  7. Interestingly, most of the patients (127|85.2%) had TARE as first-line therapy. This is unusual considering more than 40% did not have PVTT. Please comment.
  8. 11 patients were BCLC A: why they were treated with TARE?
  9. 1 patient was in stage D: same question.
  10. 5% patients had metastases. Again, this is quite unusual. Please comment.
  11. How long was the follow-up? I think that probably you have here a too short follow-up to make a solid message. Please add this datum and comment.
  12. I believe that the response (yes vs. no; type of …) to TARE should be added in the multivariate analysis to see if this factor was associated or not with survival.

Author Response

  1. All the abbreviations, starting from the abstract, should be spelled out.
  • Thank you for the comments. Corrections have been made so that any abbreviation starting from the abstracts are spelled out.

  1. Not only the exclusion criteria but also the inclusion criteria should be detailed.
  • Yes we agree and the inclusion criteria as below have been revised.
  • The inclusion criteria were patients with clinical or histological diagnosis of HCC ac-cording to HCC guideline, Child-pugh class A/B, bilirubin <2.5 mg/dL and AST/ALT <5â…¹upper limit of normal.

  1. Reading the introduction, it seems that apart from systemic therapy TAE or TARE are the only possible treatments for intermediate and advanced HCC patients. Please note that also surgery in selected cases of multinodular and/or macrovascular invasive HCC may be associated with adequate survival. The literature is replete of these data from surgical series, both from the Eastern and from the Western world.
  • Thank you for the important comment. We have added the surgical series as written below under “Introduction”.
  • Liver resection is not recommended for the intermediate stage HCC according to the BCLC classification, but in the last decades advances in surgical techniques have expanded in-dication for liver resection including the intermediate stage HCCs. There are many studies reporting improved survival in patients treated with resection over TACE in terms of survival benefit and early resection could avoid TACE refractoriness. However, for those who are not suitable to resection due to portal hypertension, TARE could be another treatment option.

  1. In this sense, TARE should be selected in those patients considered unresectable and MDT discussion should be warranted.
  • Thank you for the comment. Yes we have emphasized that TARE should be selected in those patients considered unresectable and requires MDT under “Discussion”.

  1. I found interesting the use of different HCC classifications/guidelines: from the Korea Cancer Study Group, to the BCLC, to the CLIP. This is quite unusual. Please comment.
  • I agree that too many stages are included in the analysis and this has no significant clinical implication and rather confusing. Tumor stages other than BCLC have been removed and other tumor factors such as size, number, tumor burden and are included for analysis instead.

  1. How many patients treated with TARE were excluded because of technical issues, such as shunt?
  • In our study, 19 patients were excluded from TARE treatment after performing shunt study due to hepato-pulmonary or gastrointestinal shunts.

  1. Interestingly, most of the patients (127|85.2%) had TARE as first-line therapy. This is unusual considering more than 40% did not have PVTT. Please comment.
  • Thank you for the comments. The number of patients who were treated with TARE as first-line has been changed after going over the data again and we agree that TARE is preferred to TACE when there is PVTT.
  • In our study, the reason for selecting TARE as first treatment option despite absence of major PVT was due to large tumor size at the time of diagnosis. Among 149 patient, 108 patient had tumors measuring >5cm for the largest one and 26% of these patients had tumors >10cm.
  • Another reason for selecting TARE as first treatment option was due to multiple tumors with largest tumor size (<5cm) but with infiltrative morphology involving major portal vein (n=3).
  • TARE was also performed as a mean of radiosegmentectomy for tumors measuring <5cm without major vessel invasion (n=15).
  • Since TARE is known to cause less pain than TACE for larger tumors, has more evidence for HCCs with major PVT and may have radiosegmentectomy effect, TARE was performed with wider indication in our real life study.
  • In addition to the above mentioned reasons, approximately 9 patients eligible for both TACE and TARE were treated with TACE due to the patient preference.

  1. 11 patients were BCLC A: why they were treated with TARE?
  • Thank you for your critical comments. We agree that TARE was initially indicated for patients with more advanced HCC, not suitable for TACE or for those who have fail to TACE. However, our study included a wider range of HCC stages as evolution of TARE has led to new applications in earlier-stage HCCs.
  • After going over the data, 36 patients were classified as BCLC A. Among these patients, 8 patients had single HCC with tumor size <5cm, 25 patients had single HCC sized 5-10cm and 1 patient had single HCC measuring >10cm. Resection could be another treatment option but TARE was performed as 71% of these patients were older than 60 years old and 76% of these patients were older than 75 years. Furthermore patients’ preference was another reason.
  • There were two patients with multiple HCCs <3cm. One of these patients showed 3 HCCs in segment 4, 5/6 and 7 with portal hypertension (mild ascites and esophageal varices) making resection inappropriate while another patient also had bilobar HCCs and TARE was performed as a mean of radio-segmentectomy in this patient as the small HCC in hepatic dome was not easily approached with RFA.
  • There are growing number of studies suggesting that TARE can be applied to target less-advanced HCCs and can induce complete necrosis in small (<3cm) tumor.
  • TARE also showed similar outcomes to those of ablation and surgical resection. Furthermore, location of tumor and likelihood of needle track seeding, and arterioportal fistula affected opting TARE as first line treatment option than RFA.

  1. 1 patient was in stage D: same question.
  • The patient was staged D due to misclassification to CTP C following high bilirubin level after TACE which returned to normal range during the follow-up. This patient was finally classified to CTP B and BCLC A.

  1. 5% patients had metastases. Again, this is quite unusual. Please comment.
  • Metastasis was observed in 27 patients as stated in Table 1.
  • Most of the metastasis were lymph nodes and lung single or oligometastasis. These patients had multiple or huge HCC that required local control followed by systemic treatment as the patients were ECOG 1 patients with good liver function and were able to tolerate TARE.
  • Therefore following TARE, sorafenib was prescribed in 10 patients (8 lung; 2 lymph node metastasis) radiotherapy and sorafenib in 5 patients (4 bone, 1 aortocaval lymph node metastasis) and TACE in 5 patients while 7 patients had best supportive care as they refused further treatment. The effect of following treatments were analyzed.

  1. How long was the follow-up? I think that probably you have here a too short follow-up to make a solid message. Please add this datum and comment.
  • The median follow-up duration was 16.9 months (range 1-139 months). The final state (death or alive) of all patients were obtained at the time of revision and the analysis was performed after updating the data.

  1. I believe that the response (yes vs. no; type of …) to TARE should be added in the multivariate analysis to see if this factor was associated or not with survival.
  • We appreciate your valuable comment. We have added the response in univariate and multivariate analysis to observe whether presence of response predicted better OS and PFS.
  • Tumor response remained as independent predictor of OS and PFS and tumor size was related to tumor response.

Round 2

Reviewer 1 Report

I would like to congratulate the authors on their work

Reviewer 2 Report

Dear Authors, I have read again this version of your manuscript that now has been improved following reviewers' comments.